# Neighbor Similarity and Multimodal Alignment based Product Recommendation Study

**Zhiqiang Zhang**[1,2,3]    **Yongqiang Jiang**[4]    **Qian Gao**[*1,2,3]    **Zhipeng Wang**[1,2,3]

[1]Key Laboratory of Computing Power Network and information Security, Ministry of Eaucation, Shandong Computer Science Center, Qilu University of Technology (Shandong Academy of Sciences), Jinan, China

[2]Shandong Engineering Research Center of Big Data Applied Technology, Faculty of Computer Science and Technology, Qilu University of Technology (Shandong Academy of Sciences), Jinan, China

[3]Shandong Provincial Key Laboratory of Computer Networks, Shandong Fundamental Research Center for Computer Science, Jinan, China

[4]China Telecom Digital Intelligence Techonology Co, Jinan, China

## Abstract

Existing multimodal recommendation research still faces some challenges, such as not being able to fully mine the implicit relevance information of neighbor nodes, and the unreasonable weight allocation to imbalanced nodes. To address the aforementioned challenges, this paper introduces a new multimodal recommendation model called NS-MAR+. Specifically, the model firstly constructs a neighbor similarity graph convolutional network to capture the implicit relevance information and reasonably assigns the attention weights through the graph attention mechanism. Secondly, the model introduces a modal alignment and fusion mechanism by using a multilayer perceptron (MLP) to map image and text features into a shared space for comparison and fusion. In addition, the model constructs a user co-interaction graph and an item semantic graph based on the original information and performs graph convolution operations to enhance the preference information of users and items, and to better capture the interactions and internal features between users and items. Finally, MLP is employed to aggregate user and item representations and predict personalized recommendation rankings. To validate the experiment's efficacy, this paper compares with several leading multimodal recommendation models on public datasets with a performance improvement of 1% to 3%. The experimental outcomes indicate that the model in this paper has good superiority and accuracy in multimodal recommendation tasks.

## 1   INTRODUCTION

In the era of increasingly rich and diverse information on the Web, recommender systems are widely used in various fields, aiming to provide users with more personalized and accurate recommendation services [Smith and Linden, 2017]. For example, the online shopping e-commerce platform in life, on which users can browse and purchase various commodities. The platform has rich user behavior data as well as multimodal information about the commodities. Collaborative filtering-based approaches usually only consider the direct interaction between users and commodities, while ignoring the multimodal information and the implicit relationship between users and commodities. Content-based recommendation models mainly rely on the content features of commodities for recommendation. However, the user's purchase decision may be affected by a variety of complex factors, including the user's emotion, the multimodal information of the commodity, etc. Traditional recommender methods (e.g., Collaborative Filtering [He et al., 2017]) mainly capture user preferences based on historical behavior and interaction data of users and items, but this approach tends to perform poorly in real-world datasets because the interactions between users and items are usually sparse.

To address the issue of data sparsity, researchers have introduced multimodal information (e.g., text and images) into recommendation systems [Wu et al., 2021], thereby augmenting user and item representations to improve recommendation accuracy and diversity. For example, the VBPR [He and McAuley, 2015] uses a hybrid representation of visual features and user ratings to improve recommendation accuracy through a Bayesian personalized ranking algorithm. The CKE [Zhang et al., 2016] incorporates cross-modal knowledge into a matrix factorization (MF) [Koren et al., 2009] framework. GRCN [Yinwei et al., 2021] eliminates false-positive interactions by using multimodal features. EWMIGCN [Liu et al., 2023] uses emotion-weighted graph convolutional network for personalized prediction, constructs interaction graph by extracting multimodal in-

---

*Corresponding author: Qian Gao (gq@qlu.edu.cn)

formation through parallel CNN, and propagates the information using EW-BGCN to predict user preferences in an inner product manner. FREEDOM [Zhou, 2022] freezes the item-item graph and denoises the user-item interaction graph simultaneously for multimodal recommendation. LGMRec [Guo et al., 2023] jointly models users' local and global interests by combining local and global graph learning to improve recommendation accuracy and reliability. The model of [Mu and Wu, 2023] improves the accuracy of rating prediction by mining latent features and alleviates the data sparsity problem to improve the performance of the recommender system. Simultaneously, despite the fact that multimodal recommender systems have achieved some success, they still face some challenges.

First of all, the current research suffers from the problem of failing to fully explore the association relationship between neighboring nodes when dealing with implicit correlation and assigning equal weights to nodes with different attention levels. To address these issues, this paper constructs a neighbor similarity graph convolutional network using the graph attention mechanism for multimodal recommendation tasks. Specifically, in the literature [Song et al., 2020], we find that the significance of item i to user u can be expressed through the robustness of the correlation between i and all the neighboring items of user u. Therefore, this paper creates a neighbor node similarity function for calculating the similarity coefficient between neighbor item or user nodes of a given target node. This similarity coefficient is used in the aggregation process in graph convolutional networks to assign different weights to neighbor items or users. By stacking multiple neighbor similarity graph convolutional layers, this paper is able to capture implicit correlation information between high-order neighbor nodes. Moreover, similar to LightGCN [He et al., 2020], the present structure removes feature transformations and nonlinear activations, maintaining its simplicity and interpretability.

Next, after extracting multimodal information, how to effectively fuse information from different modalities to accurately capture key features and potential relevance while avoiding information loss and distortion is the key issue faced. Modal alignment is a key technology to this problem, which ensures that correct connections and interactions are established between modalities from different sources during multimodal data processing. Explicit and implicit alignment methods are two common modal alignment strategies. For example, the Dynamic Time Warping (DTW) [Tapaswi et al., 2015] method is used to measure the similarity between two time series and find the best match between them, which realizes explicit alignment. CLIP [Radford et al., 2021] realizes implicit alignment by comparative learning of large-scale text and images. However, in practice, the performance of implicit alignment methods is highly dependent on the size and quality of the training data, and the internal decision-making process lacks transparency. In

contrast, explicit alignment methods are less capable of modeling complex nonlinear relationships and lack flexibility. In contrast, modal alignment using a multilayer perceptron has strong nonlinear modeling capabilities, flexibility, and better generalization ability to handle modal changes and transitions not encountered during training. Therefore, in this paper, by using the MLP for modal alignment, the correspondence between different modalities is explicitly considered in the shared hidden layer, which can more accurately capture the key features, optimize the information fusion, and effectively improve the performance and accuracy of the multimodal recommender system.

Finally, since the relationships between users and items are diverse, relying solely on extracted multimodal user and item features usually fails to adequately capture the complex relationships between users and items, and the processed modal features may produce ambiguous situations. Therefore, in this paper, we construct user co-interaction graph and item semantic graph based on raw information, from which we learn the potential relationship between users and items to supplement and enhance the user and item features, which further improves the accuracy and personalization of recommendation.

In summary, this paper proposes a novel multi-modal recommendation model called NSMAR+. The model is able to synthesize user behavioral data, multimodal information about commodities, and implicit relationships in order to provide more accurate and personalized recommendation results. It not only captures the user's direct preferences, but also digs deeper into the complex associations between users and commodities, thus providing a better shopping experience for users. The contribution of this paper can be summarized as follows:

- This paper constructs a novel structure of the neighbor similarity graph convolutional network and applies it to the realm of multi-modal recommendation. It takes into account the implicit correlations between neighbor nodes, capable of extracting high-order hidden information among neighbor nodes, and assigns weights to each node accordingly, providing more accurate and comprehensive inputs for subsequent tasks.

- This paper employs a multilayer perceptron for modal alignment, using MLP to process unimodal features of images and text before modal fusion, and mapping the features of both modalities to a shared feature space for alignment. The aligned features will have the same dimensions and be located in the same feature space, thereby achieving modal alignment. This method compensates for missing information in feature extraction, enhances modal correlation modeling, and improves data accuracy.

- This paper learns user co-interaction preference fea-

tures and item semantic features from user co-interaction graphs and item semantic graphs constructed based on raw information, and integrates them with the fused multimodal information to further enhance user preference features and improve the accuracy of the recommendation system.

- This paper performed experiments on three distinct public datasets from Amazon, namely Baby, Sports, and Clothing. Compared to advanced models, the resulting experimental results demonstrate the excellence of the proposed model in this paper.

## 2 METHODS

This section furnishes a comprehensive overview of the framework and constituents of the proposed NSMAR+ model, as illustrated in Figure 1. The innovations of this study primarily focus on the first three modules. Firstly, neighbor similarity graph convolutional network (Figure 2 shows the detailed structure of the network) are constructed in the unimodal feature extraction module to capture user preferences, item representations, and implicit correlation information among neighbors of each node specific to each modality, and different weights are assigned to neighbor nodes with different levels of attention, as depicted by the red dashed box 'a' in Figure 1. Next, the extracted unimodal features are modally aligned using MLP in the multimodal alignment and fusion module, and features are fused using the splicing method to solve the problems of information loss and insufficient modal correlation modeling, so as to inductively learn the multimodal representations of the users and the items as shown in the red dashed box b in Figure 1. (The blue dashed box 'd' is the NSMAR+ model experiment, and the NSMAR model experiment replaces this part with the learnable modal alignment matrix.) Subsequently, the user co-interaction graph and item semantic graph based on the original information are constructed in the graph learning module, which are used to capture the co-interaction relationship between users and the semantic relationship between items, so as to supplement the enhancement of the user and preference features as shown by the red dashed box c in Figure 1 shown. Finally, in the prediction module, a multi-layer perceptron is utilized to aggregate the final user and item features. The preference scores are then computed for ranking recommendations.

### 2.1 PREPARATION

Given a set of M user nodes $U = \{u_1, u_2, \ldots, u_M\}$ and a set of N item nodes $I = \{i_1, i_2, \ldots, i_N\}$, this paper uses symbols u and o as examples of user nodes, and i and j represent examples of item nodes. By constructing the user-item bipartite graph $G = \{u, i, \varepsilon\}$ to model user interaction

behavior, where u and i denote the sets of users and items, respectively, and $\varepsilon$ represents the set of historical interactions. In addition to the interaction relationships between users and items, each item also has multi-modal content information denoted as $m = \{v, t\}$, where v and t denote visual and textual features, respectively. The multimodal feature representation of item i is denoted as $x_i^m \in R^{d_m}$, where $d_m$ represents the feature dimension of modality m. Furthermore, the IDs of users and items are mapped to embedding vectors $u^{(0)} \in R^d$ and $i^{(0)} \in R^d$, where d represents the dimension of embeddings. To construct the embedding layer for message propagation, this paper creates trainable embedding lookup tables.

$$E_u = \left[ u_1^{(0)}, \quad u_2^{(0)}, \quad \ldots, \quad u_M^{(0)} \right], \qquad (1)$$

$$E_i = \left[ i_1^{(0)}, \quad i_2^{(0)}, \quad \ldots, \quad i_N^{(0)} \right]. \qquad (2)$$

where M, N respectively denote the count of users and items.

### 2.2 SINGLE-MODAL FEATURE EXTRACTION MODULE

In the realm of multi-modal recommendation systems, a common issue is the insufficient exploration and extraction of similarity information among neighboring nodes, along with the rational allocation of corresponding weights to different nodes. This may lead to an inaccurate capture of crucial features and user preferences across different modalities, thereby impacting the precision and effectiveness of recommendations. To tackle this challenge, we propose a novel approach by constructing a neighbor similarity graph convolutional network on the user-item bipartite graph of each modality. This method considers not only the user preferences and item features within each modality but also emphasizes the implicit similarity relationships among neighboring nodes, assigning appropriate weights accordingly. In comparison to previous methods, this approach comprehensively explores and extracts similarity information among neighboring nodes, mitigating issues of information loss and distortion. It effectively models the complex inter-modal relationships, captures latent correlations among modalities, enhances the model's generalization performance, and thereby accurately captures key features and user preferences across different modalities. The network structure is illustrated in Figure 2.

Additionally, in this network architecture, a novel neighbor node similarity function is introduced to determine the similarity coefficients between user and item neighbor nodes. The structure diagram of this similarity function is illustrated in Figure 3. In this context, ReLU activation function is employed here to ensure that the output result is always non-negative. And the paper utilizes Pearson similarity [Gadekula et al., 2019] within this function to measure the linearity between two continuous variables. It ranges

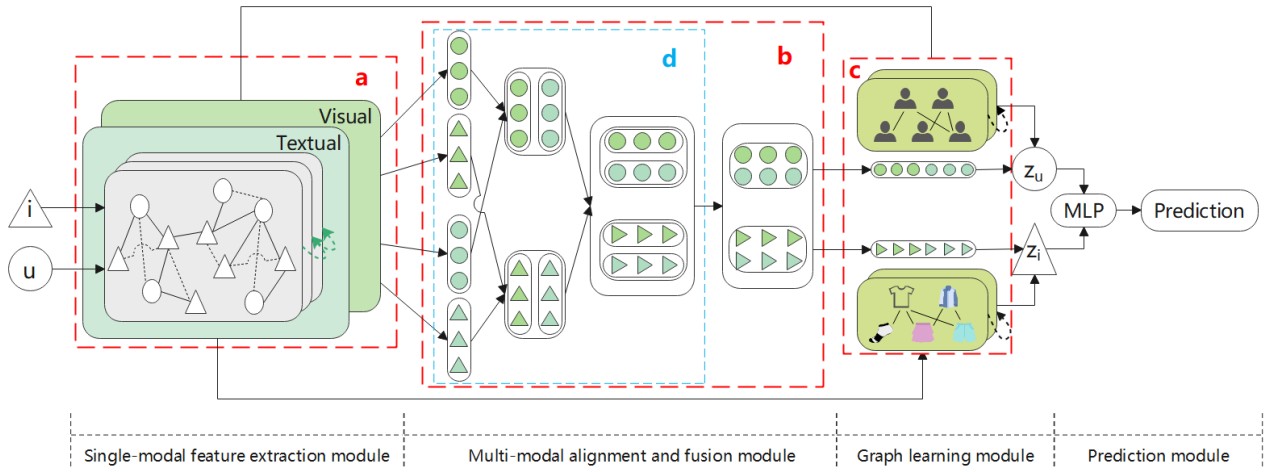

Figure 1: The overall framework of the proposed NSMAR+.

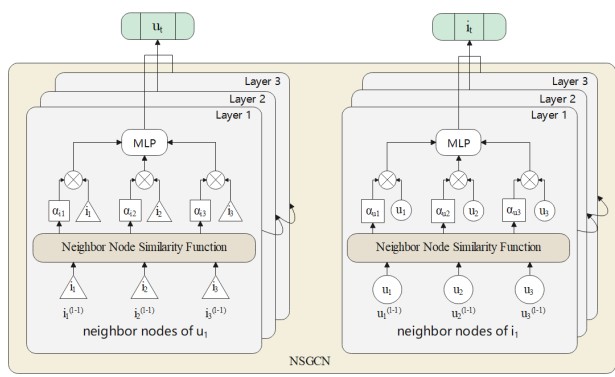

Figure 2: Neighbor similarity graph convolutional network.

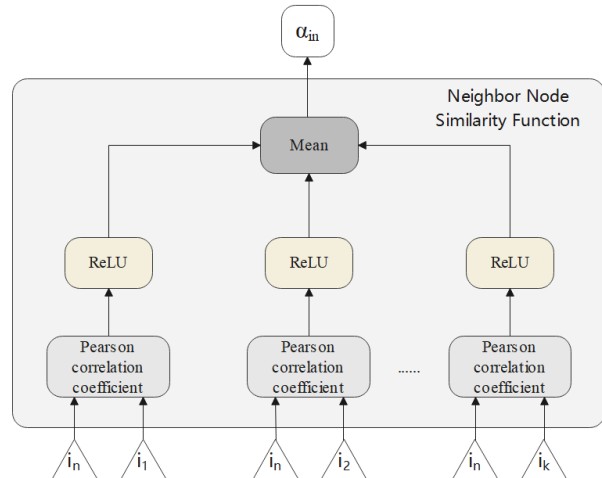

Figure 3: Structure diagram of the neighbor node similarity function.

from -1 to 1, indicating the degree of correlation between the two variables. Its advantage lies in being relatively robust to outliers, less susceptible to extreme values, and widely applicable. At the same time, to simplify the computation process, this paper performs an averaging operation on the results of each pairwise similarity function calculation. In summary, the paper obtains the neighbor node similarity coefficients for each node in the message propagation process:

$$\alpha_i^{(l-1)} = \frac{1}{|N_u|} \sum_{j \in N_u} \text{ReLU}(r(i^{(l-1)}, j^{(l-1)})), \quad (3)$$

$$\alpha_u^{(l-1)} = \frac{1}{|N_i|} \sum_{o \in N_i} \text{ReLU}(r(u^{(l-1)}, o^{(l-1)})). \quad (4)$$

where $i^{(l-1)}$, $j^{(l-1)}$ represent the embeddings of different neighboring items for user u in the $l-1$ th layer, while $u^{(l-1)}$, $o^{(l-1)}$ represent the embeddings of different neighboring users for item i in the $l-1$ th layer. $N_u$, $N_i$ are the neighboring items of user u and neighboring users of item i in the graph, respectively; $|N_u|$, $|N_i|$ represent the number

of neighboring items for user u and the number of neighboring users for item i in the graph. $\alpha_i^{(l-1)}$ represents the similarity coefficient of the neighbor item i aggregated to user u in the $l-1$ th layer, and $\alpha_u^{(l-1)}$ represents the similarity coefficient of the neighbor user u aggregated to item i in the $l-1$ th layer.

In summary, this paper obtains the modal feature representations of users and items for the l th layer as follows:

$$u^{(l)} = \frac{1}{\sqrt{|N_u|}} \text{MLP} \left( \sum_{i \in N_u} \alpha_i^{(l-1)} \cdot i^{(l-1)} \right), \quad (5)$$

$$i^{(l)} = \frac{1}{\sqrt{|N_i|}} \text{MLP} \left( \sum_{u \in N_i} \alpha_u^{(l-1)} \cdot u^{(l-1)} \right). \quad (6)$$

To solve the issue of potentially large embedding norms

after aggregation, this paper adopts scaling method based on $\frac{1}{\sqrt{|N_u|}}$ or $\frac{1}{\sqrt{|N_i|}}$, which can effectively keeps the embedding norms within an acceptable range, ensuring the stability and effectiveness of the model.

Following l layers of data propagation, this paper obtains the final representations of users and items for a single modality by stacking multiple neighbor similarity graph convolutional layers as follows:

$$u_m = \sum_{l=0}^{l-1} u^{(l)}, \qquad (7)$$

$$i_m = \sum_{l=0}^{l-1} i^{(l)}. \qquad (8)$$

## 2.3 MULTIMODAL ALIGNMENT AND FUSION MODULE

### 2.3.1 Modal alignment

Considering that there are complementary information or missing information problems between different modal data, resulting in the extracted features may not be able to fully express the multidimensional information of users and items. Therefore, in this paper, by employing a multilayer perceptron for modal alignment, text and image features are processed separately using MLP, and then the processed features are passed to the shared layer for modal alignment. This approach not only reduces information loss and distortion, but also enhances the inter-modal correlation modeling effect and significantly improves the recommendation accuracy, as illustrated in the blue dashed box 'd' in Figure 1.

In this module, this paper maps the features of different modalities to the shared space through MLP for modal alignment, and obtains the embedded representations of the aligned users and items (taking items as an example):

$$i_t^* = \text{MLP}\,(i_t), \qquad (9)$$

$$i_v^* = \text{MLP}\,(i_v). \qquad (10)$$

### 2.3.2 Multimodal fusion

To enhance the precision of the multimodal recommendation system, the key lies in effectively integrating multimodal information. During the fusion process, we integrate the aligned modal embedding information to construct a comprehensive multimodal representation. Specifically, for the extracted modal features, we adopt a stitching fusion mechanism for multimodal fusion, as illustrated by the red dashed box 'b' in Figure 1. This mechanism directly splices the

features of different modalities, preserves the original information of each modality, enhances data integrity, simplifies the feature fusion process, and improves the efficiency of multimodal fusion.

$$u_{mf} = \text{concat}(u_v^*, u_t^*), \qquad (11)$$

$$i_{mf} = \text{concat}(i_v^*, i_t^*). \qquad (12)$$

## 2.4 GRAPH LEARNING MODULE

### 2.4.1 User co-interaction graph

Following the principle that users who have interacted frequently with the same items usually share similar multimodal preferences, if two users frequently interact with the same items, an edge is considered to exist between them. Therefore, this paper constructs a user co-interaction graph $G_1 = \{U, e\}$ based on the initial embedding information, where nodes represent users and edges represent co-interaction relationships between users. Here, $e = \{(u, o)|u, o \in U\}$ denotes the set of edges between co-occurring nodes in $G_1$. In addition, this paper adopts the Softmax-weighted aggregation method [Velickovic et al., 2017] during graph propagation. The formula is as follows:

$$u_{co}^{l+1} = \sum_{o \in N_u'} \left( \frac{\exp(e_{u,o})}{\sum_{o' \in N_u'} \exp(e_{u,o'})} \right) u. \qquad (13)$$

where $u_{co}^{l+1}$ represents the user co-interaction representation learned in the $l + 1$ th layer of the user co-interaction graph $G_1$. $e_{u,o}$ represents the count of frequently co-occurred items between user node u and o. $N_u'$ represents the neighboring item nodes of user u in $G_1$.

### 2.4.2 Item semantic graph

In the initial embedding information, there is also a layer of latent associative information between items, namely, item semantic information. This paper creates a semantic graph $G_2 = \{I, e\}$ based on the initial embedding information, where items are represented as nodes, and the associations between items are represented by edges. $e = \{(i, j)|i, j \in I\}$ represents the set of edges in graph $G_2$ connecting semantically similar nodes of items. Similarly, this paper employs the Softmax-weighted aggregation method during graph propagation on the semantic graph, the formula is as follows:

$$i_{se}^{l+1} = \sum_{j \in N_i'} \left( \frac{\exp(e_{i,j})}{\sum_{j' \in N_i'} \exp(e_{i,j'})} \right) i. \qquad (14)$$

where $i_{se}^{l+1}$ represents the item semantic representation learned at the $l + 1$ th layer in the item semantic graph $G_2$. $e_{i,j}$ signifies the weight assigned to the edge connecting item nodes i and j. $N_i'$ denotes the neighboring user nodes of item i in $G_2$.

## 2.5 PREDICTION MODULE

### 2.5.1 Integration of user and item representations

After processing user and item features through multiple modules and components discussed earlier, this paper obtained various representations for users and items. These representations are subsequently merged to acquire the ultimate representations for users and items.

$$Z_u^* = u_{mf}^l + u_{co}^l, \tag{15}$$

$$Z_i^* = i_{mf}^l + i_{se}^l. \tag{16}$$

where $Z_u^*$ and $Z_i^*$ represent the final representations for user u and item i, respectively.

Then, this paper employs a multi-layer perceptron to fuse the final user features and item features and compute the preference score of the user for the item, as shown below:

$$y_{ui} = \text{MLP}(Z_u^*, Z_i^*). \tag{17}$$

where $y_{ui}$ denotes the preference score of the user for the target item.

### 2.5.2 Optimization

To optimize the arguments of NSMAR+, this paper employs the Bayesian Personalized Ranking (BPR) loss [Rendle et al., 2009] to rank the scores of user-item pairs. And during the training process, this paper constructs a set of triplets M and utilizes the integrated final embeddings of users and items to formulate the objective function of the BPR loss. The loss function $L_{\text{rec}}$ is formulated as follows:

$$M = \{(u, i, j) | (u, i) \in \varepsilon, (u, j) \notin \varepsilon\}, \tag{18}$$

$$L_{rec} = \sum_{(u,i,j) \in M} - \ln \sigma(y_{ui} - y_{uj}) + \lambda ||\Theta||^2. \tag{19}$$

where the triplet set M constitutes the complete training data, $\sigma()$ denotes the sigmoid function, $\lambda$ represents the regularization weight coefficient, and $\Theta$ represents the model arguments.

## 3 EXPERIMENTS

### 3.1 EXPERIMENTAL SETUP

#### 3.1.1 Datasets

To access the capability of leveraging multimodal data for recommendations in real-world scenarios, this study conducted experiments and performance assessments on three datasets from Amazon: Baby, Sports, and Clothing. These datasets include both textual descriptions and images for

Table 1: Dataset Statistics.

| Dataset | Users | Items | Interactions |
|---------|-------|-------|--------------|
| Baby | 19,445 | 7,050 | 160,792 |
| Sports | 35,598 | 18,357 | 296,337 |
| Clothing | 39,387 | 23,033 | 278,677 |

each item, allowing them to be treated as text and image features, respectively. Each purchase record is treated as a user-item interaction. To ensure the quality of the datasets, the study conducted preprocessing, retaining only those users and items with at least 3 interactions. For text and image features, the study employed pre-trained sentence-transformers [Reimers and Gurevych, 2019] to extract 384-dimensional text features and utilized previously published 4096-dimensional image features [Zhang et al., 2021]. Table 1 provides detailed information about these three datasets.

#### 3.1.2 Evaluation indicators and parameter setting

This study employed a random partitioning method to allocate the dataset into three subsets: 80% for training, 10% for validation, and 10% for testing. In the experiments, the embedding size was configured as 64, and the batch size was set to 2048. And according to the hyperparameter experiments in Section 4.4, in this paper, we set the maximum sampled neighbor count $K$ and the regularization weight $\lambda$ to 60 and 1e-3, respectively. Additionally, Recall and Normalized Discounted Cumulative Gain (NDCG) were chosen as evaluation metrics for appraising the top-K recommendation performance of the proposed model. The study used the Xavier method [Glorot and Bengio, 2010] for initializing all model embeddings and employed the Adam optimizer [Kingma and Ba, 2014] for model optimization. And, this study also used analysis of variance (ANOVA) to assess the importance of improvement. Before performing ANOVA, we first ensured the normality of the data and variance comparison. Then, we compared the performance of the experimental and control groups using different recommendation algorithms and found that the experimental group significantly outperformed the control group in both NDCG and Recall metrics, with p-values less than the pre-determined level of significance (0.05), thus indicating that these differences were statistically significant.

#### 3.1.3 Baseline

To showcase the effectiveness of NSMAR+, this study conducted comparisons with the following baseline models:

- **LightGCN** [He et al., 2020] is a graph convolutional network simplified by discarding feature transformations and non-linear activation modules.

- **LATTICE** [Zhang et al., 2021] introduces item-item graphs based on multimodality and aggregates the information of each modality to obtain the potential item semantic graphs.

- **MMGCN** [wei Wei et al., 2019] builds a fully connected network with a long and short-term memory network to extract modal features.

- **DualGNN** [Wang et al., 2023] adapts LightGCN for multi-modal recommendation and constructs a user co-interaction graph, enabling graph learning to merge representations of neighbors in user-related graphs.

- **MGCN** [Peng et al., 2023] incorporates behavioral information and designs a behavioral-aware fuser and self-supervised assisted task to comprehensively model user preferences.

- **DRAGON** [Zhou et al., 2023] utilizes LightGCN to extract unimodal features and over constructs homogeneous and heterogeneous graphs to learn dual representations of users and items.

## 3.2 PERFORMANCE COMPARISON

To assess the effectiveness of the NSMAR+ proposed in this paper, we conduct experimental comparisons with advanced baselines, and the comparison results are shown in Table 2.

All GCN-based CF methods consistently outperform traditional MF methods. This is because graph convolution operations are better at integrating multimodal data, addressing cold-start issues, and effectively modeling the complex network structure between users and items, thereby improving the performance of recommendation systems.

Multimodal-based methods are usually superior to GCN-based methods because the former can integrate information from more data sources and enhance robustness and accuracy. For example, MGCN and LATTICE outperform LightGCN.However, in the case of small datasets or emphasizing graph structure modeling, GCN-based methods still have some advantages. Taking the Clothing, Baby and Sports datasets as examples, DualGNN works better on the Clothing dataset but not on the other two datasets, which may be related to the fact that the clothing dataset is richer in multimodal information. In addition, MMGCN does not perform as well as LightGCN on all datasets, probably due to the introduction of information that is not related to user preferences. The NSMAR+ method proposed in this paper, which synthesizes higher-order collaborative filtering and multimodal features and consistently maintains high

performance levels in all aspects, likewise supports the conclusions of this analysis.

Compared to LightGCN, LATTICE and MMGCN, NSMAR+ employs neighbor similarity graph convolutional networks to extract personalized features and implicit information among neighbor nodes, and assigns weights to neighbor nodes with different attention levels, which in turn improves recommendation accuracy. Compared with DualGNN, MGCN, and DRAGON, NSMAR+ introduces a graph learning module, which complements the augmented preference features by constructing user co-interaction graphs based on raw information, item semantic graphs capturing co-interaction relationships between users, semantic relationships between items, and providing a more comprehensive user and item representation. And compared with NSMAR, NSMAR+ performs modal alignment through MLP before modal fusion, which has better nonlinear modeling and generalization capabilities, makes up for the shortcomings of explicit and implicit alignment methods in terms of flexibility and data dependence, etc., and can more accurately capture the key features and optimize the fusion of information, so as to solve the problems of information loss and insufficient modal relevance modeling more effectively. In addition, NSMAR+ uses a multilayer perceptron to fuse final user features and item features and computes preference scores to rank recommendations. Overall, NSMAR+ achieves 1% to 3% improvement on all three datasets, indicating that the model proposed in this paper has some positive effects and innovation.

## 3.3 ABLATION STUDY

This study performed ablation experiments to assess the influence of different factors on the proposed model. In this paper, the following variants are constructed by removing specific components from the NSMAR+ model:

- NSMAR: It replaces MLP alignment with explicit modal alignment methods.
- NSMAR+/NC: It removes the neighbor similarity graph convolutional network.
- NSMAR+/MA: It removes the modal alignment.
- NSMAR+/GUI: It removes the user co-interaction graph and the item semantic graph.
- NSMAR+/T: It removes textual information.
- NSMAR+/V: It removes visual information.

Based on the observations in Figure 4, it is evident that the different variant operations have a significant impact on the model. Specifically, the variant NSMAR+/NC decreases the model performance after removing the neighborhood similarity graph convolutional network, which is due to the fact that this component helps in capturing the implicit correlation information and assigning the weights rationally.

Table 2: Overall performance Comparison.

| Dataset | Metric | SLIM | FISM | LightGCN | LATTICE | MMGCN | DualGNN | MGCN | DRAGON | NSMAR | NSMAR+ | Imp |
|---|---|---|---|---|---|---|---|---|---|---|---|---|
| Baby | Recall@10 | 0.0440 | 0.0451 | 0.0479 | 0.0547 | 0.0376 | 0.0448 | 0.0620 | 0.0662 | **0.0669** | 0.0672 | 1.51% |
| | Recall@20 | 0.0693 | 0.0709 | 0.0754 | 0.0850 | 0.0614 | 0.0716 | 0.0964 | 0.1021 | **0.1034** | 0.1037 | 1.57% |
| | NDCG@10 | 0.0229 | 0.0241 | 0.0257 | 0.0292 | 0.0200 | 0.024 | 0.0339 | 0.0345 | **0.0352** | 0.0354 | 2.60% |
| | NDCG@20 | 0.0285 | 0.0296 | 0.0328 | 0.0370 | 0.0261 | 0.0309 | 0.0427 | 0.0435 | **0.0445** | 0.0448 | 2.98% |
| Sports | Recall@10 | 0.0515 | 0.0521 | 0.0569 | 0.0620 | 0.0371 | 0.0568 | 0.0729 | 0.0749 | **0.0758** | 0.0760 | 1.47% |
| | Recall@20 | 0.0827 | 0.0836 | 0.0864 | 0.0953 | 0.0602 | 0.0859 | 0.1106 | 0.1124 | **0.1135** | 0.1139 | 1.33% |
| | NDCG@10 | 0.0276 | 0.0284 | 0.0311 | 0.0335 | 0.0193 | 0.0310 | 0.0397 | 0.0403 | **0.0410** | 0.0415 | 2.97% |
| | NDCG@20 | 0.0323 | 0.0330 | 0.0384 | 0.0421 | 0.0253 | 0.0385 | 0.0496 | 0.0500 | **0.0510** | 0.0514 | 2.80% |
| Clothing | Recall@10 | 0.0318 | 0.0323 | 0.0361 | 0.0492 | 0.0205 | 0.0454 | 0.0641 | 0.0650 | **0.0665** | 0.0667 | 2.61% |
| | Recall@20 | 0.0507 | 0.0506 | 0.0544 | 0.0733 | 0.0330 | 0.0683 | 0.0954 | 0.0957 | **0.0972** | 0.0978 | 2.19% |
| | NDCG@10 | 0.0163 | 0.0166 | 0.0197 | 0.0268 | 0.0109 | 0.0241 | 0.0347 | 0.0357 | **0.0367** | 0.0369 | 3.36% |
| | NDCG@20 | 0.0220 | 0.0218 | 0.0243 | 0.0330 | 0.0141 | 0.0299 | 0.0428 | 0.0435 | **0.0441** | 0.0446 | 2.52% |

Secondly, the variant NSMAR+/MA also negatively affects the performance by removing the modal alignment module, which may lead to feature mismatch during modal fusion. Similarly, the variant NSMAR+/GUI removes the user co-interaction graph and the item semantic graph, which affects the model's ability to accurately extract user preference features. This negatively affects the accuracy and degree of personalization of the recommendation algorithm, which leads to a degradation of the model performance. Finally, compared to NSMAR+, the variant NSMAR uses an explicit modal alignment method with poor nonlinear modeling and generalization capabilities, which can lead to reduced accuracy and efficiency in subsequent fusion tasks.

In addition to this, two main points can be drawn from the above observations: first, there are differences in the performance of different modalities. The results in Figure 4 show that NSMAR+/V outperforms NSMAR+/T in most cases, indicating that the textual modality usually performs better in the unimodal case. Second, the multimodal model NSMAR+ consistently outperforms the unimodal model variant, suggesting that modeling and fusion of joint visual and textual modalities better improves recommender system performance.

In summary, the experimental results indicate that the proposed model outperforms other variants. This indicates that each component of the model plays a positive role in enhancing performance.

### 3.4 HYPER-PARAMETER STUDY

#### 3.4.1 Investigating the effect of maximum sampled neighbor count (K) on model performance

In real-world recommendation scenarios, there are typically numerous users and items. Employing a message propagation method that calculates the weights of all neighbors for each node increases complexity. Therefore, In the neighbor similarity graph convolutional network, this paper adopts the Max-K sampling method, which aims to optimize the

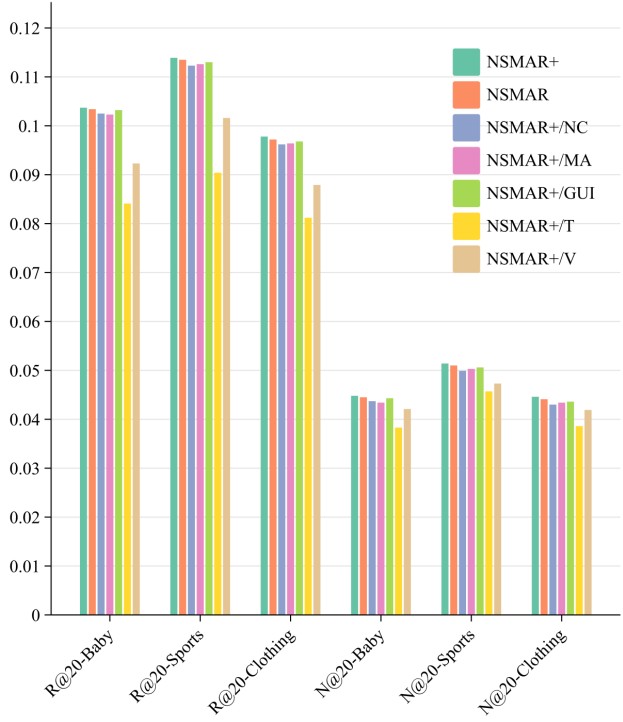

Figure 4: The effect of enhanced modules on model performance.

model training. The experimental results of this approach are shown on the left in Figure 5, which shows the variation of model performance with the maximum number of sampled neighbors, K, on different datasets. It can be observed that the model performance initially improves as K increases, but a performance difference occurs after a certain threshold is reached. On the Baby dataset, the performance peaks at K = 60 and then decreases, while on the Sports and Clothing datasets, the performance stabilizes after K = 60 and starts to decrease after K = 80. This indicates that Sports and Clothing datasets have more and denser data. The overall model performance shows a trend of increasing and then decreasing due to insufficient information when the number

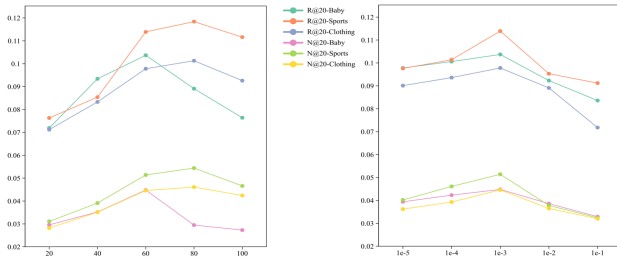

Figure 5: The effect of hyperparameters on model performance.

of neighbors is too small, and may be interfered by invalid neighbors when there are too many. Therefore, K = 60 is chosen as the maximum number of sampled neighbors to balance the training effect.

### 3.4.2 Investigating the effect of regularization weight ($\lambda$) on model performance

According to the results of the right panel in Figure 5, we can conclude that the optimal regularization weight ($\lambda$) values for the three datasets are around 1e-3. As the value of $\lambda$ gradually increases beyond 1e-3, the performance of the model significantly decreases. The reason for this decline is that excessive regularization may lead to underfitting, reducing the model's robustness and potentially causing premature convergence, negatively impacting the model's training and, consequently, its performance. Therefore, we choose the regularization weight ($\lambda$) to be 1e-3 to achieve optimal performance.

## 4 CONCLUSION AND FUTURE WORK

This paper proposes an improved multimodal recommendation model called NSMAR+. The model firstly captures the implicit correlation information between different modal neighbor nodes effectively by constructing a neighbor similarity graph convolutional network and assigns corresponding weights to it. Then, this paper performs modal alignment by using multilayer perceptron before modality to solve the problems of mismatch of modal feature information and data loss and distortion, and at the same time, reduces the information dimension and extracts potential correlation between modalities, so as to obtain more accurate and comprehensive data information. Finally, this paper constructs a user co-interaction graph and an item semantic graph to enhance the final preference features by capturing the correspondence information through graph convolution. After extensive experimental validation, the NSMAR+ model significantly improves the recommendation performance on all three datasets, verifying its effectiveness. However, this paper concludes that the model still has room for improvement. In future research, the modal fusion methods can be

further optimized to solve the problem of unreasonable allocation of modal information weights, so as to improve the interpretability and accuracy of the model.

## Acknowledgements

This work was supported by the Natural Science Foundation of Shandong Province (ZR2022MF333).

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
