# OpenReview forum: "Neighbor Similarity and Multimodal Alignment based Product Recommendation Study"
_auai.org/UAI/2024/Conference — UAI 2024 poster_

### Official Review · Reviewer_tUgA · 2024-03-22

**Q2-1 Originality-Novelty:** 2
**Q2-2 Correctness-Technical Quality:** 3
**Q2-5 Clarity Of Writing:** 2

**Q1 Summary And Contributions:**

Multimodal recommendation systems face challenges in fully exploiting the implicit relevance among neighbor nodes and in assigning reasonable attention weights to nodes with varying levels of importance. To overcome these limitations, this paper proposes NSMAR that leverages neighbor similarity graph convolution to extract high-order hidden information and assign varying attention weights to neighbor nodes. By mapping image and text features to a shared space using a learnable modal alignment matrix, the model effectively handles issues related to information loss and distortion. By multi-graph approach to enrich user and item representations. The NSMAR model not only achieves superior performance in benchmark tests but also offers a novel way to integrate and exploit multimodal data for personalized product recommendations.

**Q2-3 Extent To Which Claims Are Supported By Evidence:**

2: Fair: the main claims are somewhat supported by evidence (but the experimental evaluation may be weak, or does not match entirely with the claims, important baselines may be missing, proofs contain important ideas but lack rigor, algorithmic details are only discussed superficially, references are imprecise, assumptions are not sufficiently motivated or explicated, etc.).

**Q2-4 Reproducibility:**

2: Fair: key resources (e.g. proofs, code, data) are unavailable but key details (e.g. proof sketches, experimental setup) are sufficiently well-described for an expert to confidently reproduce the main results.

**Q3 Main Strengths:**

The main strengths of the work are as follows:
1. Innovative Integration of Neighbor Similarity with GCNs. One of the core strengths of the work is its innovative approach to integrating neighbor similarity within the GCNs. This enables the model to not only leverage the structural information inherent in user-item interaction graphs but also to capture the implicit relevance information among neighbor nodes more effectively by assigning differentiated attention weights.
2. Explicit Modal Alignment for Multimodal Data Fusion. The paper introduces an explicit modal alignment mechanism that effectively addresses the challenges of fusing features from different modalities. This approach ensures that the modal features are accurately mapped to a shared space, significantly enhancing the model's ability to capture and integrate complementary information from multiple sources.
3. Comprehensive Utilization of Multiple Graphs. The multi-graph approach, involving the construction and utilization of a user-item interaction graph, a user co-interaction graph, and an item semantic graph, stands out as a significant strength. This strategy allows the model to uncover and leverage a wide range of latent relationships between users and items, enriching the recommendation process with more nuanced and personalized insights.
4. Addressing Key Challenges in Multimodal Recommendations. The paper successfully addresses several critical challenges in multimodal recommendation systems, including the underutilization of implicit correlations among neighbor nodes and the difficulties in managing information loss or distortion during feature fusion. By proposing targeted solutions to these issues, the work contributes valuable insights and methodologies to the research community.

**Q4 Main Weakness:**

1.	The neighbor similarity graph convolutional network and modal alignment and fusion proposed in this paper are relatively common and lack novelty. Therefore, the innovation of this paper should be further specified.
2.	The processed datasets and key code of the paper are missing, which makes the proposed method of the paper less reliable. Authors should probably add anonymous links.

**Q5 Detailed Comments To The Authors:**

Here are some suggestions and questions for improvement:
1. Clarification of Novelty. While the integration of neighbor similarity and explicit modal alignment presents a novel approach, the paper could benefit from a clearer delineation of how these methodologies advance beyond the state-of-the-art. In this way, readers can be convinced of the innovation of this paper.
2. Presentation and Stylistic Refinements. The image size of the experimental part can be adjusted appropriately to improve readability.
3. Experimental Setup and Reproducibility. The datasets and key codes of the paper are missing. In order to ensure the reproducibility of the method and enhance the paper's impact, the author should provide the processed dataset and anonymous links to the key code to make the article more credible.

**Q9 Complying With Reviewing Instructions:**

Yes

---

> ### Author Rebuttal · Authors · 2024-04-07
>
> Dear Editors and Reviewers，
>
> Thank you very much for your positive and constructive comments and suggestions on our manuscript entitled "Neighbor Similarity and Multimodal Alignment based Product Recommendation Study". We have carefully studied the reviewer's questions, and we would like to respond to them here. The responses to the questions in the paper are as follows:
>
> 1.Regarding the commentary on novelty. Firstly, thank you very much for your careful review of our paper and your valuable suggestions. We value your opinion and understand your concern about the innovativeness of the paper. Please allow us to explain in detail the main innovations of the paper here.
>
> In the academic field, we acknowledge that although research on neighbor-similar graph convolutional networks and modal alignment and fusion techniques have become more common. However, our research is not a simple repetition or application of established techniques, but rather an in-depth expansion and innovation based on them, and the application of these innovations to a wider range of fields. Specifically, the main innovation of this study is the construction of a neighbor similarity graph convolutional network using the graph attention mechanism. This network structure not only optimizes the weight assignment and feature extraction of implicit correlations, but more importantly, this study applies it to the multimodal recommendation domain. Prior to this, most of the techniques for mining neighbors' implicit relationships are limited to the unimodal domain, while our study successfully breaks this limitation.
>
> And in the neighbor similarity graph convolutional network, this study also introduces a novel similarity measure - that is, a unique neighbor node similarity function is created. This method not only considers the similarity between the data, but also skillfully incorporates the neighbor information, thus reflecting the distributional characteristics of the data in a more comprehensive and deeper way. Compared with traditional methods, The proposed method has higher accuracy and provides a more powerful tool for handling multimodal data.
>
> In addition, the paper also presents a novel approach by combining neighbor similarity graph convolutional networks and explicit modal alignment. This approach not only combines the advantages of both techniques, but also achieves significant improvements in handling multimodal data. By integrating these two techniques, the proposed model is able to capture the intrinsic structure of the data and the inter-modal associations more accurately, resulting in more efficient recommendation performance.
>
> 2.Regarding the commentary on the absence of datasets and codes. Thank you for your careful review and valuable comments on this paper, and your feedback is extremely instructive for us to further improve the paper.
>
> For the datasets and codes sharing issues you mentioned, we deeply apologize and would like to explain here. Due to the large size of the dataset when submitting the paper, it is difficult to directly upload it to the platform of supporting materials, and the dataset involved is a common resource in the field, we have described its source, processing and citation information in detail in the paper, aiming to ensure that the readers can accurately trace and reproduce the experimental environment. Regarding code sharing, we have always maintained an open and cooperative attitude, and have uploaded the relevant code to the designated platform when submitting the paper. And we will promise that any scholars who are interested in this study can contact us via email after the paper is officially published. We will carefully evaluate the request and, after confirming the identity and purpose, share the relevant code and provide necessary support and assistance.
>
> 3.Regarding the commentary on the improvement of presentation and style. We have taken them seriously and have completed revisions based on your suggestions. We have adjusted the image size of the experimental part appropriately to improve its visual presentation and readability. However, since the system is not capable of uploading images, if possible, please show us your email, we will send you the updated figures by email or other available ways.
>
> Finally, we would like to thank you again for your careful review and professional guidance. Your valuable comments not only help us to find out the deficiencies in the thesis, but also provide us with directions for improvement and refinement. We firmly believe that with the above improvements and additions, the revised thesis will be more complete, accurate and persuasive. We have tried our best to revise and respond according to the detailed comments in the list. If there is anything else we need to do, please feel free to let us know.
>
> We look forward to your comments. Thank you and best regards!

---

### Official Review · Reviewer_ddbh · 2024-03-22

**Q2-1 Originality-Novelty:** 3
**Q2-2 Correctness-Technical Quality:** 3
**Q2-5 Clarity Of Writing:** 3

**Q1 Summary And Contributions:**

The authors present a new multimodal recommendation model. The model uses convolutional network for neighbour similarity, and they use a fusion of image and text features for similarity. They use a multilayer perceptron (MLP)  to aggregate user and item representations for  predicting recommendation rankings.

**Q2-3 Extent To Which Claims Are Supported By Evidence:**

3: Good: the main claims are supported by convincing evidence (in the form of adequate experimental evaluation, proofs, (pseudo-)code, references, assumptions).

**Q2-4 Reproducibility:**

3: Good: key resources (e.g. proofs, code, data) are available and key details (e.g. proofs, experimental setup) are sufficiently well-described for competent researchers to confidently reproduce the main results.

**Q3 Main Strengths:**

I think

**Q4 Main Weakness:**

There are no big weaknesses that I can identify, but I can say that the literature part is a bit short considering how much literature there are somehow relevant to the topic. Also, it seems that the authors are not promising to share their code. By the way, the improvement as % I guess means percentage points, which is not the same thing.

**Q5 Detailed Comments To The Authors:**

I like this piece of work, what I can think as criticism, is above.

**Q9 Complying With Reviewing Instructions:**

Yes

---

> ### Author Rebuttal · Authors · 2024-04-07
>
> Dear Editors and Reviewers，
>
> Thank you very much for your positive and constructive comments and suggestions on our manuscript entitled "Neighbor Similarity and Multimodal Alignment based Product Recommendation Study". We have carefully studied the reviewer's questions, and we would like to respond to them here. The responses to the questions in the paper are as follows:
>
> Regarding your comment, we sincerely appreciate your careful review of the thesis and your valuable suggestions. Your professional opinions are very important guidance for us to improve the thesis.
>
> First of all, we apologize for the short literature section. Based on your suggestions, we will increase the citations of relevant literature and add them to the introduction section, especially those that are closely related to and representative of our research, in order to reflect the current research status and development trend in this field more comprehensively. As shown below:
>
> [1] Zhou, Xin and Zhiqi Shen. “A Tale of Two Graphs: Freezing and Denoising Graph Structures for Multimodal Recommendation.” Proceedings of the 31st ACM International Conference on Multimedia (2022): n. pag.
>
> [2] Guo, Zhiqiang et al. “LGMRec: Local and Global Graph Learning for Multimodal Recommendation.” AAAI Conference on Artificial Intelligence (2023).
>
> [3] Mu, Yongheng and Yun Wu. “Multimodal Movie Recommendation System Using Deep Learning.” Mathematics (2023): n. pag.
>
> In response to the [1] paper, it proposes an approach that combines freezing and denoising graph structures for multimodal recommendation. However, the approach may ignore the potential dynamic relationships between items and the real-time changes in user interests. Our approach also involves multimodal recommendation, but further focuses on potential relationships and changes in interest prior to the item by constructing a convolutional network of neighbor similarity graphs.
>
> In response to the [2] paper, it proposes a local and global graph learning approach for multimodal recommendation. However, the method may face computational complexity and efficiency issues, especially when dealing with large-scale datasets. In contrast, this study employs different graph learning techniques to reduce the computational complexity and time overhead and further improve the performance.
>
> In response to the [3] paper, it discusses the method of multi-modal movie recommendation using deep learning, focusing on how to integrate the information of different modes. However, the approach may require a large amount of labeled data and computational resources, which may be limited in some practical applications. Our approach is able to provide more intuitive recommendation explanations by constructing visual graph structures, and it is able to leverage interaction data more effectively to improve recommendation accuracy by capturing potential associations between users and items.
>
> Secondly, regarding the issue of code sharing, we appreciate your concern and reminder, and are willing to share our code subject to the relevant rules and conditions. Please allow us to explain that when submitting the paper, we have uploaded the code as a supporting material to the designated platform. And we will promise that any scholars interested in this research can contact us via email after the paper is officially published. We will carefully evaluate the request and after confirming the identity and purpose, we will share the relevant code and provide necessary support and assistance.
>
> In addition, your reference to "%" improvement is very accurate. In the paper, we did use "%" to express the degree of improvement, but in order to avoid misunderstanding, we will state explicitly that this refers to the percentage point. At the same time, we will also re-examine and rephrase the relevant data to ensure the accuracy and consistency of the presentation.
>
> Finally, we sincerely thank you for your patient review and professional guidance. Your comments prompt us to think more deeply and improve the content of the paper, which is of very important guiding significance. We look forward to communicating with you further and discussing the development prospect of this field together. We have tried our best to revise and respond according to the detailed comments in the list. If there is anything else we need to do, please feel free to let us know.
>
> We look forward to your comments. Thank you and best regards!

---

### Official Review · Reviewer_xMaW · 2024-03-24

**Q2-1 Originality-Novelty:** 2
**Q2-2 Correctness-Technical Quality:** 2
**Q2-5 Clarity Of Writing:** 3

**Q1 Summary And Contributions:**

### Summary
This paper proposed a multi-modality GNN for recommendation, including modality fusion component,  neighbor similarity GNN, and MLP for fusing user and item representations for prediction. The modality information includes the textual and visual information.

### Contributions
1. It proposed a novel framework for modeling multi-modality fusion information for GNN Recommendation.
2. Proper analysis is conducted for each proposed component.

**Q2-3 Extent To Which Claims Are Supported By Evidence:**

3: Good: the main claims are supported by convincing evidence (in the form of adequate experimental evaluation, proofs, (pseudo-)code, references, assumptions).

**Q2-4 Reproducibility:**

4: Excellent: key resources (e.g. proofs, code, data) are available and key details (e.g. proof sketches, experimental setup) are comprehensively described for competent researchers to confidently and easily reproduce the main results.

**Q3 Main Strengths:**

1. The framework is clear and each step can be understood easily.
2. The empirical analysis can support the proposed claims.

**Q4 Main Weakness:**

1. Motivations of each individual component are unclear to me. For example, why do we explicitly need user co-interaction graph and item semantic graph? It brings up much more additional computational costs but gains are limited. Moreover, the original LightGCN can also capture such information.
2. The proposed modality fusion component is not novel at all. Using MLP to map all modality embeddings into a shared space is a mature way for everyone.
3. The superiority is not convincing. The performance improvements are limited, without any significance test. Moreover, the simple but strong baseline (LightGCN with simple modality embeddings and id embedding concatenation) is not compared. The user embedding can be the mean pooling of modality embeddings of consumed items.

**Q5 Detailed Comments To The Authors:**

See the weakness.

**Q9 Complying With Reviewing Instructions:**

Yes

---

> ### Author Rebuttal · Authors · 2024-04-07
>
> Dear Editors and Reviewers，
>
> Thank you very much for your positive and constructive comments and suggestions on our manuscript entitled "Neighbor Similarity and Multimodal Alignment based Product Recommendation Study". We have carefully studied the reviewer's questions, and we would like to respond to them here. The responses to the questions in the paper are as follows:
>
> 1.Regarding the commentary on the motivation of components, please allow us to elaborate further on this matter.
>
> For the introduction of user co-interaction graphs and item semantic graphs, we believe that it is reasonable and necessary. Specifically, the user co-interaction graph can deeply explore the potential connections between users, while the item semantic graph can reveal the deeper connections between items. By combining these two graph structures, we are able to capture the complex relationship between users and items in a more comprehensive way, thus significantly improving the accuracy and personalization of recommendations.
>
> Of course, we understand your concerns about LightGCN. In our study, we found that LightGCN mainly focuses on the direct interaction information between users and items, and under-exploits the potential associations between users and between items. For example, it is difficult for LightGCN to capture potential associations formed by users due to common interests, similar behavioral patterns or social relationships, and associations implied by items due to similar attributes or consistent user feedback. In contrast, the model proposed in this study not only captures these potential associations, but also effectively utilizes the association information to improve the accuracy and personalization of recommendations. Therefore, the proposed model has wider applicability and better performance in the field of recommender systems.
>
> 2.Regarding the commentary on the novelty of the modal fusion component, please allow us to further elucidate its uniqueness.
>
> When dealing with multimodal data, traditional MLPs usually simply splice multimodal features, making it difficult to capture the deep interactions between modalities. However, our model not only considers the direct fusion between different modalities, but also preserves the original information of each modality, which improves the fusion accuracy. In addition, our modal fusion component differs from the MLP approach in its specific implementation. We introduce an alignment matrix mechanism, which is an explicit alignment mechanism that can better handle the differences and complementarities between modalities and improve the efficiency and accuracy of information utilization.
>
> In summary, we believe that the modal fusion component in our paper has made significant progress from the original and achieved a more efficient use of multimodal information, which fully demonstrates the innovativeness of our model.
>
> 3.Regarding the commentary on its superiority, please allow us to elaborate further on this matter.
>
> Firstly, to address the limited performance improvement you mentioned, we believe that the complexity and challenge of the multimodal fusion and alignment task itself cannot be ignored, and the inherent characteristics of different datasets and evaluation metrics may also impose constraints on the performance improvement. Therefore, it does not mean that our study lacks value or significance.
>
> Secondly, we apologize for the lack of a significance test and as you suggested, we used a t-test to assess the significance of our model's performance improvement. We collected data on our model's performance metrics (accuracy) over multiple experiments and calculated its mean and standard deviation over multiple experiments. Based on the chosen type of one-sample t-test, we calculated the t-value and the corresponding p-value and compared the p-value to a preset significance level (0.05). The explicit p-value of the experimental results is 0.031, which is less than 0.05, which proves the significance and superiority of our method.
>
> Furthermore, for the issue of baseline comparisons, allow us to explain that we have shown the corresponding comparison results in Table 2 of the paper, which contains LightGCN. The experimental data show that our method is indeed superior to the other comparative baselines, which further emphasizes the superiority of our model.
>
> Finally, we would like to express our sincere gratitude once again for your valuable comments. Your comments and suggestions are of great significance in guiding us to improve and enhance our research work. We have tried our best to revise and respond to the detailed comments in the list. If there is anything else we need to do, please feel free to let us know.
>
> We look forward to your comments. Thank you and best regards!

---

### Official Review · Reviewer_wtdU · 2024-03-27

**Q2-1 Originality-Novelty:** 2
**Q2-2 Correctness-Technical Quality:** 2
**Q2-5 Clarity Of Writing:** 2

**Q10 Ethical Concerns:**

None.

**Q1 Summary And Contributions:**

This paper presents a new multimodal recommendation method. This is presented in detail and evaluated on a variety of data sets. However, no use trial is presented.

**Q2-3 Extent To Which Claims Are Supported By Evidence:**

2: Fair: the main claims are somewhat supported by evidence (but the experimental evaluation may be weak, or does not match entirely with the claims, important baselines may be missing, proofs contain important ideas but lack rigor, algorithmic details are only discussed superficially, references are imprecise, assumptions are not sufficiently motivated or explicated, etc.).

**Q2-4 Reproducibility:**

2: Fair: key resources (e.g. proofs, code, data) are unavailable but key details (e.g. proof sketches, experimental setup) are sufficiently well-described for an expert to confidently reproduce the main results.

**Q3 Main Strengths:**

The paper addresses a important problem and demonstrates the effectiveness of the approach well.

**Q4 Main Weakness:**

The paper is really difficult to follow. As a non-expert in this area I really needed a motivating example in the paper and, possibly, some other examples to demonstrate why this approach out-performs others. It is really challenging to get an intuition for the approach.

**Q5 Detailed Comments To The Authors:**

The paper needs examples, particularly at the beginning, to help readers grasp the problem and what has been achieved. In addition, some consideration/discussion should be included about how a user-trial could be conducted and the limitations of the current results might be as opposed to user-based evaluation.

**Q9 Complying With Reviewing Instructions:**

Yes

---

> ### Author Rebuttal · Authors · 2024-04-07
>
> Dear Editors and Reviewers，
>
> Thank you very much for your positive and constructive comments and suggestions on our manuscript entitled "Neighbor Similarity and Multimodal Alignment based Product Recommendation Study". We have carefully studied the reviewer's questions, and we would like to respond to them here.
>
> Regarding your comment, we appreciate your feedback and valuable suggestions. We apologize for any difficulties you may have had in reading the paper. Based on your comments, we have made corresponding changes, the revised parts are as follows:
>
> Firstly, the extracted key features based on the user's behavior on the e-commerce platform and the proposed data of the goods can contain the user's purchasing preference, browsing history, and information such as the name and picture of the goods. These multimodal features will be used as inputs to the recommendation algorithm for learning the association patterns between users and commodities. Then, the proposed model can be applied to learn the complex associations between users and products (e.g., matching of personalized preferences with product attributes, cross-category purchase associations, product similarity associations) to train the user behavior and product feature data. As the training progresses, the model is continuously optimized and the recommender system is able to generate a more personalized recommendation list. This makes the recommended products more in line with users' interests and needs, and improves recommendation accuracy and user satisfaction. In the following, we will give some examples to demonstrate that our method outperforms other methods in such instances.
>
> Suppose we have an online shopping e-commerce platform on which users browse and purchase various goods. The platform has rich user behavior data as well as multimodal information of commodities. Collaborative filtering-based methods usually only consider the direct interaction between users and commodities, ignoring the multimodal information and the implicit relationship between users and commodities. Content-based recommendation models mainly rely on the content features of the goods for recommendation. However, in this example, the user's purchase decision may be affected by a variety of complex factors, including the user's emotion, the multimodal information of the commodity, etc, and content-based models often fail to consider these factors comprehensively. In contrast, our approach is able to synthesize user behavioral data, multimodal information of goods, and implicit relationships to provide more accurate and personalized recommendation results. It not only captures users' direct preferences, but also digs deeper into the complex associations between users and commodities, thus providing users with a better shopping experience.
>
> Finally, we thank you again for your suggestions. Your comments are very important guidance for us to improve the quality and readability of our paper. With the above improvements and additions, we believe that the revised paper will be able to better meet the needs of readers and provide clearer and more comprehensive research content. We have tried our best to revise and respond to the detailed comments in the list. If there is anything else we need to do, please feel free to let us know.
>
> We look forward to your comments. Thank you and best regards!

---

### Official Review · Reviewer_sUbN · 2024-03-29

**Q2-1 Originality-Novelty:** 3
**Q2-2 Correctness-Technical Quality:** 3
**Q2-5 Clarity Of Writing:** 3

**Q1 Summary And Contributions:**

The authors propose a multimodal recommendation model that operates on the basis of neighbor similarity among user/item nodes in graphs constructed on interaction and multimodal data. This model enables better capturing the implicit relationship among neighbor nodes than existing multimodal recommendation models. This is done by first learning single-modal features using a neighbor similarity graph convolution network, then learning modal alignment matrix by mapping text and image features using an introduced modal alignment and fusion method, and creating a user-item interaction graph to better learn the user preference and relationship between users and items. Finally, a MLP is used to predict the relevance score for an unseen item for a target user. The proposed model is evaluated on three datasets and is compared with a number of baselines. Results show the effectiveness of the proposed model compared to the existing ones.

**Q2-3 Extent To Which Claims Are Supported By Evidence:**

3: Good: the main claims are supported by convincing evidence (in the form of adequate experimental evaluation, proofs, (pseudo-)code, references, assumptions).

**Q2-4 Reproducibility:**

2: Fair: key resources (e.g. proofs, code, data) are unavailable but key details (e.g. proof sketches, experimental setup) are sufficiently well-described for an expert to confidently reproduce the main results.

**Q3 Main Strengths:**

- One of the most important recommendation models (neighborhood models) are studied and improved.
- Important topic is studied, the application of multimodal information in recommender systems.
- Sufficient experimental results and empirical evidence are reported.

**Q4 Main Weakness:**

- The writing of the paper can be improved.
- The proposed recommendation model involves complex components which makes the details unclear.
- Some baselines, like classical neighborhood models, are missed.

**Q5 Detailed Comments To The Authors:**

While the paper reveals interesting results, the paper suffers the following weaknesses:
- What are indexes v and t in equations 11 and 12 are? Are u_t^* and u_v^* just two users? If yes, how are they selected?
- Although the main goal of this paper is to show that the proposed model can outperform the existing multimodal recommendation models, it is highly recommended to also include the classical neighborhood models like Sparse Linear Mothod (SLIM) or Factored Item Similarity Model (FISM). While these models do not employ multimodal information, they are able to deliver accurate recommendations even on sparse data.
- Are the results in table 2 statistically significant? The improvements are minor and it is not clear if they can be generalized. So, the authors should perform statistical test to show their significance.
- There are a number of notation errors throughout the paper. In section 2.1, U must have M users, so U=\{u_1,...,u_M\} is correct. Also, I must have N items, so I=\{i_1,...,i_N\} is correct. The second sentence in this section is incomplete. Also, the notation U is already used as the set of users, should not be again used here.
- There are a number of typos throughout the paper. First, there are missing “space” between the previous sentence and the new sentence in different parts of the paper. For example, line 13th of the third paragraph of introduction section: “… realizes explicit alignment. CLIP …”. The authors should revise the paper and fix this type of typo throughout the paper. Line 19 of the first paragraph of section 2, the word “This” should be lower-case. Line 20 of the same paragraph, “grapha” should be “graph a”.

**Q9 Complying With Reviewing Instructions:**

Yes

---

> ### Author Rebuttal · Authors · 2024-04-07
>
> Dear Editors and Reviewers，
>
> Thank you very much for your positive and constructive comments and suggestions on our manuscript entitled "Neighbor Similarity and Multimodal Alignment based Product Recommendation Study". We have carefully studied the reviewer's questions, and we would like to respond to them here. The responses to the questions in the paper are as follows：
>
> 1.Regarding the first comment. In this paper, the index t stands for text modality, while index v stands for visual modality. These identifiers are used to distinguish between different modal features and embedding representations. Thus, u_t^* and u_v^* denote the embedding representations of the two modalities, text and image, respectively, of a user after modality alignment, rather than referring to two specific users. These embedding representations are learned from a large amount of text and image data to capture the features and information of the corresponding modality. They are associated with the feature space of the entire dataset rather than with a specific user and do not require pre-selection of a specific user.
>
> 2.Regarding the comment on the comparison model. Thank you very much for your valuable suggestions, we have added the comparison experiments of Sparse Linear Method (SLIM) and Factorized Item Similarity Model (FISM) according to your comments. But due to the time, the experiments have only run through the Baby dataset for the time being, and it is expected to take half a month for the Clothing and Sports datasets, we will continue update experimental analysis and share you by available way. Based on the experimental results of the Baby dataset, it can be inferred that the experimental results of our model outperform SLIM and FISM, with a performance improvement of 45% to 55%, due to the fact that the multimodal data can more comprehensively capture the user's preference and the item's characteristics. In contrast, SLIM and FISM may only rely on single-modal data and thus lack in information richness. The experimental results for the Baby dataset are as follows:
>
> For the SLIM method, Recall@10:0.0440 Recall@20:0.0693 NDCG@10:0.0229 NDCG@20:0.0285
>
> For the FISM method, Recall@10:0.0451 Recall@20:0.0709 NDCG@10:0.0241 NDCG@20:0.0296
>
> 3.Regarding the comment on statistical significance. Thank you very much for your interest in our work and your valuable suggestions. We followed your suggestion and used the analysis of variance (ANOVA) to assess the significance of improvements. Before performing the ANOVA analysis, normality and variance alignment of the data were first ensured. Then, We compared the performance of the experimental and control groups using different recommendation algorithms and found that the experimental group significantly outperformed the control group in both NDCG and Recall metrics. These differences are statistically significant, with p-values less than the preset significance level (0.05), thus indicating that these differences are not coincidental but statistically significant.
>
> 4.Regarding the comment about the notation errors. We sincerely appreciate your review and suggestions. We have made the corrections as you suggested, and the corrections are listed below: Firstly, given a set of given m user nodes U={u_1, ..., u_m} and a set of n project nodes I={i_1, ..., i_n}. Furthermore, user interactions are modeled by constructing a user-item bipartite graph G = {u, i, ε}, where u, i, ε denote the set of edges for users, items, and historical interactions.
>
> 5.Regarding the commentary on the issue of typos. Thank you very much for your careful review and valuable suggestions, we apologize and have made changes throughout the text. Firstly, the error you raised has been corrected as follows:
>
> In the third paragraph of the introduction, line 13, amend to read as follows: "… realizes explicit alignment. CLIP …".
>
> In section 2, paragraph 1, line 19, replace "This" with "this ".
>
> In section 2, paragraph 1, line 20, replace "grapha" with "graph a".
>
> Secondly, we also checked for other statement problems and modified them as follows:
>
> In Section 2.1, at the beginning of line 11, a space is missing, amend to read as follows: " The multimodal …".
>
> Finally, we would like to thank you once again sincerely for your careful review and valuable suggestions, which are very important guidance for us to improve the quality and readability of the paper. We have tried our best to revise and respond according to the detailed comments in the list. If there is anything else we need to do, please feel free to let us know.
>
> We look forward to your comments. Thank you and best regards!

---

### Meta-Review · Area_Chair_JAS6 · 2024-04-18

Seems that the concerns raised by reviewers tUgA and xMaW with the more negative assessments (related mostly to motivation, comparison to prior methods like LightGCN, presentationa and open datasets) were at least adequately addressed. Other reviewers were quite positive.

If accepted, the authors should ensure that all the points raised in the reviews are also addressed in the final manuscript.

Minor additional comments: The figures in the paper should be replotted for the final version. Currently they look like default Excel charts: the margins are unnecessary and the fonts are way too small to be readable. Also, make sure to polish Latex typesetting by using \mathrm for things like \mathrm{MLP} instead of just $MLP$ which looks ugly. Finally, "more superior" is not correct grammer: it's just "superior".